



# Prediction of Hysteretic Matric Potential Dynamics Using Artificial Intelligence: Application of Autoencoder Neural Networks

*Nedal Aqel[1*], Lea Reusser[1,2,3], Stephan Margreth[2], Andrea Carminati[1], and Peter Lehmann[1]*

*[1]Physics of Soils and Terrestrial Ecosystems , ETH Zurich, Switzerland*
*[2]Office for the Environment, canton of Solothurn, Switzerland*
*[3]now at the Swiss Academy of Sciences, Forum Landscape, Alps, Parks (FoLAP), Switzerland*

*corresponding author, nedal.aqel@usys.ethz.ch, Universitätstrasse 16, 8092 Zurich, Switzerland

**Abstract**

**Information on soil water potential is essential to assess soil moisture state, to prevent soil compaction in weak soils, and to optimize crop management. In lack of direct measurements, the soil water potential values must be deduced from soil water content dynamics that can be monitored at plot scale or obtained at larger scale from remote sensing information. Because the relationship between water content and soil water potential in natural field soils is highly ambiguous, the prediction of soil water potential from water content data is a big challenge. The hysteretic relationship observed in nine soil profiles in the region of Solothurn (Switzerland) is not a simple function of texture or wetting and drainage cycles but depends on seasonal patterns that may be related to soil structural dynamics. Because the physical mechanisms governing seasonal hysteresis are unclear, we developed a deep neural network model that predicts water potential changes using rainfall, potential evapotranspiration, and water content time series as inputs. To adapt the model for multiple locations, we incorporated a Deep Autoencoder Neural Network as a classifier. The autoencoder compresses the water content time series into a site-specific feature that is highly representative of the underlying water content dynamics of each site and quantifies the similarity of dynamic patterns. By adding the Autoencoder's output as an additional input and training the neural network model with three stations located in three major classes founded by the autoencoder, we predict matric potential for other sites. This method has the potential to deduce the dynamics of matric potential from water content data (including satellite data) despite strong seasonal effects that cannot be captured by standard methods.**



## 1. Introduction


The soil water characteristics curve SWC relates the matric potential (MP) and water content (WC) and
is the key physical property to quantify soil water dynamics (Tuller & Or, 2023). The SWC (also
denoted as soil water retention curve or pressure-saturation relationship) depends on both soil texture
and structure and differs with soil types and soil textural classes (Rawls, et al., 2003; Shwetha & Varija,
2015). The SWC contains information on the pore size distribution and allows the assessment of flow
and transport properties for different hydration states (Rostami, et al., 2015; Menon, et al., 2020). To
provide a complete characterization of the actual soil moisture state and flow regimes, information on
both the matric potential and the water content must be specified. Information on volumetric water
content is needed to assess the free storage capacity, optimize water management, and to formulate
mass balance. The matric water potential is a component of the total and hydraulic soil water potential
and determines the water flow in direction of decreasing water potential to achieve equilibrium with its
surroundings (Ma, et al., 2022). The matric potential is also of particular interest to assess mechanical
stability of a soil (Holthusen, et al., 2010; Lu, et al., 2010). The capillary and adsorptive forces expressed
with the matric potential define the unsaturated soil strength mitigating soil compaction by heavy
machinery in construction work, farming, and forestry (Smith, et al., 2001). For example, matric
potential thresholds are defined in various regions of Switzerland to prevent mechanical damage and
regulate the maximum load linked to factors like soil type, texture, and vehicle impact (Bundesamt für
Energiewirtschaft, 1997). Other important potential thresholds are the wilting point and the field
capacity, characterizing the plant available water (Gupta, et al., 2023).
It would be optimal to determine the soil moisture status relative to these potential thresholds based on
information of water content using the SWC, without direct measurement of the matric potential. In that
case, matric potential dynamics could be deduced from remote sensing water content data that are
available at various scales. However, the application of this procedure is limited by two effects. Firstly,
under saturated conditions, the water potential can change without modifying the volumetric water
content. The transition of conditions with negative water potential within the capillary fringe to positive
pressures below a water table is crucial for the triggering of landslides (Gallipoli, et al., 2003). Secondly,



the SWC under field conditions is often an ambiguous relationship between potential and water content
due to hysteretic and dynamic effect as will be discussed next.
The SWC is typically measured in the lab as series of equilibrium states obtained during drainage, with
one water content value assigned to the applied pressure. The results of such small-scale experiments
are not sensitive to structural pores that can be found at the field scale (Romero-Ruiz, et al., 2018) and
can thus be expressed as function of basic soil properties (texture, bulk density, content of organic
material) using pedotransfer functions (PTF; Zuo & He, 2021). Because these PTFs ignore the effects
of soil structures including macropores and cracks (Basile, et al., 2019) and are trained with data from
small samples with artificially high initial saturation conditions, their applicability to model dynamic
processes in the field is limited. Another limitation is the underlying assumption of an unambiguous
relationship between water content and matric potential (and hydraulic conductivity). In all land surface
models, water content is linked by an unambiguous relationship between water content and matric
potential. In reality, this relationship is highly ambiguous under field conditions as was analyzed in
detail by Hannes et al. (2016) and as we will show later in this paper as well.
Hannes et al. (2016) analyzed long-term experiments and concluded that the high variation of matric
potential values for the same water content are a result of hysteresis, dynamic effects, and structural
changes during the season. Hysteresis is related to differences in wetting and drying cycles (Capparelli
& Spolverino, 2020) as controlled by different pore structures controlling air- or water invasion and
differences in receding or advancing wetting angles (Fomin, et al., 2023). Hysteresis is often manifested
in coarse textured soils and occurs as well during slow processes. Another process resulting in an
ambiguous pressure-saturation relationship is dynamic effects with water contents that are not in
equilibrium with the quickly changing potential (Ross & Smettem, 2000). Finally, the size of structural
pores is not constant with time but changes with season, water content, and soil formation processes
(Fu, et al., 2021). The combined effect of hysteresis, non-equilibrium, and structural changes makes it
extremely challenging to deduce soil matric potential from information on water content. Also, the
implementation of these combined effects in physically-based models of unsaturated water flow is not
straightforward. As an alternative approach to physically-based models, machine learning can be





applied to simulate the complex relationship between matric potential and water content under field
conditions. In this study, we will apply a deep neural network (DNN).
Deep neural networks (DNN) have demonstrated their effectiveness as a powerful numerical tool for
resolving complex patterns. Their ability to learn from data and recognize intricate relationships makes
them valuable in various fields, including the modeling of soil water characteristics. For example, Jain,
et al. (2004) and Achieng (2019) used artificial neural network (ANN) models to predict the hysteretic
water content from observed matric potential values. However, both publications simulated lab data
under equilibrium conditions and cannot be applied for the more complex dynamic processes in the
field. In addition, the models were site-specific and needed both water content and matric potential
information for the training. Here we will apply a different DNN using an autoencoder approach. As
we will explain in the theory section, the autoencoder condenses the complexity of temporal (and
spatial) patterns into a single (or a few) number(s). The hypothesis of this study is that the autoencoder
value is a new and unique characterization of the soil moisture dynamics and can be used to predict
matric potential dynamics from observed water content data. The paper is organized as follows: in
section 2, the study sites and the basics of the deep neural network with the autoencoder approach are
presented. The results section compares the model performance of site-specific deep neural network
(DNN) and shows the possibility to build a generalized DNN using the autoencoder analysis as model
input. Limits and possible applications of the model approach are discussed in section 5.

## 2. Material and methods

In a first step, matric potential time series were simulated at nine sites in the region of Solothurn
(Switzerland) using site specific ANN model, to proof that the ANN models can predict matric potential
from water content dynamics with site specific training. In the next step, the autoencoder analysis of
water content dynamics of all sites was conducted. Finally, the site-specific ANN model was enhanced
and transformed into a multisite model by combining two deep neural networks. This transformation
allowed for a more comprehensive and versatile predictive framework of matric potential as function
of water content.



## 2.1 Study area and soil moisture data


The study area covers mainly the canton of Solothurn in Switzerland (Fig.1), and thus an area of
approximately 629 km$^2$. The climate in Solothurn is classified as oceanic climate (Cfb) according to
Koppen and Geiger climate classification, with an average yearly temperature of 9.5 °C and annual
precipitation of around 1400 mm. Approximately half of the annual precipitation in the canton
undergoes the process of evaporation (Auer, et al., 2005). During the year, the average temperature
varies by 19 °C with the highest temperature occurring in the month of July and the lowest average
temperature in January. Regarding precipitation patterns, the month of June has the highest level of
precipitation, while March stands out as the driest month. Soil moisture dynamics (see below) were
studied for the period from 2011 to 2022. For this period, climatic data were available on the data portal
of MeteoSwiss (Federal Office for Meteorology and Climatology, 2023). The data was gathered from
the closest meteorological stations to each of the nine sites in the Solothurn region.
Soil moisture data were downloaded from the 'soil monitoring network' (Bodenmessnetz; BMN, 2023)
collecting data from 65 stations distributed over eleven cantons of Switzerland. The network's primary
objective is to provide real-time soil moisture information for mitigating soil compaction. BMN also
plays a role in raising awareness among farmers and foresters about soil compaction, providing a tool
to assess the current situation and adjust the use of heavy machinery based on weather conditions. As
the network has been running since 2011, it now serves as a valuable resource by offering long-term
diverse information, including land use, precipitation amounts, and matric potential measured at various
depths (20 and 35 cm depth in most of the stations, using T8 and T32 tensiometers from METER group).
Only at nine sites that are located in the region of Solothurn, the water content was measured at 20 cm
depth (Stevens Hydra Probe). For these nine sites, daily values in volumetric water content (20 cm),
matric potential (20 cm) and precipitation values were used.
As the soil moisture decreases, water is drawn from the tensiometer, creating a negative pressure or
tension. During dry periods, cavitation may occur, causing water vaporization and air bubble formation
(Mendes & Buzzi, 2013), or tensiometers had to be refilled (Sadeghi, et al., 2020). To address these
challenges and ensure accurate data collection, various data preprocessing and filtering techniques were





implemented. These techniques involved identifying and removing outliers, systematically excluding
data points with water potential values within the problematic dry ranges and filtering out data points
with extremely low or high water content values. The study also flagged abrupt changes in volumetric
water content (VWC) and matric potential (MP) for further investigation, as these could indicate
measurement anomalies. Additionally, a thorough analysis of weekly trends in the data was conducted
to identify systematic variations over time (see Appendix A).

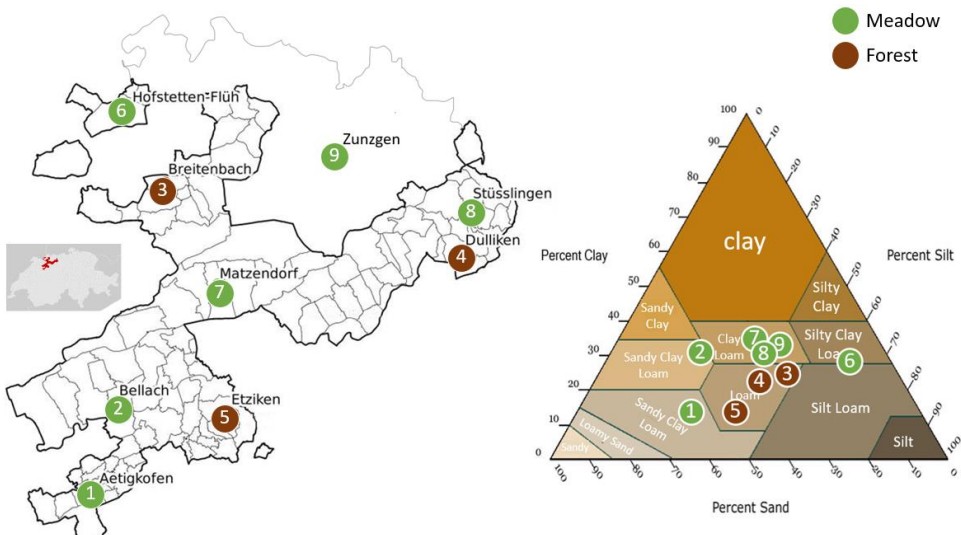


**Figure 1** Overview of the study area with site locations, soil texture, and land cover. The primary focus
is on the canton of Solothurn, outlined by the black border on the map, with an additional site from the
canton of Basel (site 9, Zunzgen). Within this region, three sites are categorized as forests, while the
remaining six sites are designated as meadows. The analyzed soil horizons (20 cm depth) of the study
area encompasses five soil textural classes as shown in the soil texture triangle.
The analyzed soil horizons of the selected locations can be assigned to five different soil textural classes
(figure 1) and two different land covers (meadow and forest). The location denoted as Matzendorf (site
#7) contains the highest clay content, whereas locations such as Aetigkofen (site #1) are predominantly
sandy. Across these nine locations, different relationships between matric potential and water content
were deduced from field data as shown in Figure 2 for two sites with low and high variations in water
content for similar potential values. To show the relevance of seasonal patterns, we differentiate
between summer (April to September) and winter period (remaining months).




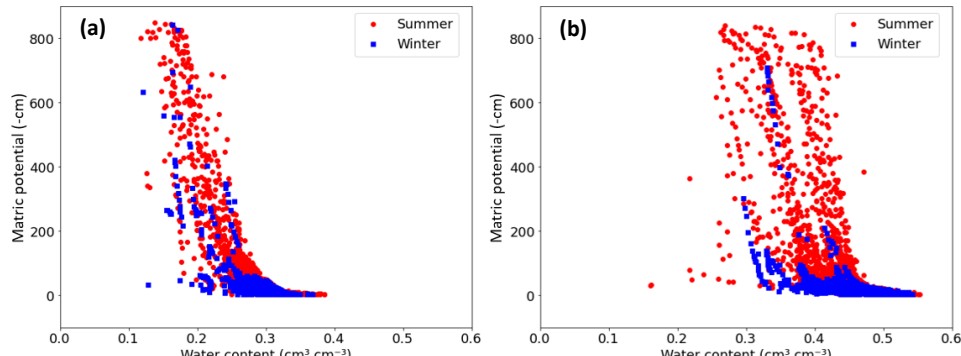


**Figure 2** Soil-Water characteristics curve (SWC) measured in the field at two sites classified into summer (April to September) and winter period (remaining months) from 2012 to 2023. **(a)** The Etziken site (site #5) shows small changes in the SWC dynamics over the years, for both the warm and cold period. **(b)** A contrasting scenario was found for the site in Bellach (site #2) that was characterized by a wide range of water content for similar potential values.

2.2 Deep neural network (DNN)
A basic artificial neural network (ANN) comprises one or two hidden interconnected layers, with each
layer tasked with the conversion of an input vector (**x**) into a hidden state vector (**h**), as described by
(Bertels & Willems, 2023). This conversion is accomplished through the utilization of a weight matrix
(**W**) and a bias vector (**b**), integrated with an activation function (denoted as "act" in eq (1)).
$h = f(x) = act(W.x + b)$ (1)
To construct a deep neural network (DNN), multiple layers (more than two hidden layers) are
interconnected to form a 'multilayer perceptron.' The training process involves finding optimal values
for the weights and biases in the network using suitable optimization techniques (Bertels & Willems,
2023). In this study, DNN was built to predict the daily MP for the nine sites. The process involved
several key steps. First, in the design of the neural network, activation functions were carefully selected
and integrated to introduce non-linearity into the model's transformations (Montesinos Lópezm, et al.,
2022). The Rectified Linear Unit (ReLU) activation function was employed to mitigate vanishing
gradient problem and enhance the model's ability to handle noisy input. The inclusion of ReLU was
motivated by considerations of computational efficiency, with some attention given to the potential
issue of "dying ReLU" (Montesinos Lópezm, et al., 2022; Lu, 2020).



Next, the neural network was structured with a total of six layers, including four hidden layers as
suggested by Achieng (2019). All layers were densely connected, fostering strong information flow
between neurons. Crucially, batch normalization was incorporated after the second hidden layer. Batch
normalization is a technique that normalizes the activations within a layer during training, which can
help mitigate issues like internal covariate shift and accelerate convergence (Ioffe, 2015). The choice
of the optimization method was the Adam optimizer, a powerful tool for training neural networks. It
adaptively adjusted learning rates, thereby optimizing the learning process, and enabling rapid
convergence while employing Mean Squared Error (MSE) as the loss function (Kingma & Ba, 2014).
To prevent overfitting by the Adam optimizer, an early stopping mechanism was implemented. This
mechanism continuously monitored the loss function for the hold out data during training, ceasing the
process if no improvement or a sudden increase was detected over a predetermined number of
consecutive epochs.
The initial deep neural networks (DNN) were configured with 4 input parameters and the daily
logarithmic scaled matric potential (MP) value as output. The input parameters consisted of
precipitation, potential evapotranspiration, measured VWC, and the weekly percentage change in
VWC. As the prediction process progressed, two major issues were identified. Firstly, the influence of
the VWC measurements on the training process was found to be predominant. Consequently, a decision
was made to increase the weight of precipitation and potential evapotranspiration in the calculation
process by incorporating three new input parameters: the weekly total precipitation and
evapotranspiration (the sum of the current day and the preceding six days), along with the difference
between these two new components. Secondly, the use of logarithmic scaled MP values was found to
be highly sensitive to data availability. Therefore, a decision was made to retrain the model using
absolute linear MP values (see Appendix B). In total, the final model was equipped with 7 input
parameters to predict the absolute linear MP values for a given location. For each site, a site-specific
DNN was built. The extent of the training data is predominantly influenced by site-specific
characteristics. For instance, sites characterized by sandy soils necessitated a shorter training duration
in contrast to sites with a higher clay content. Typically, the training dataset spanned a duration of 4 to



7 years. During this period, 70% of the data were randomly selected for training, while the remaining
30% were set aside as holdout data (Gholamy, et al., 2018). The extra years of data beyond the initial
training period were reserved for validation purposes.
2.3 Autoencoder neural network
The autoencoder, consisting of an encoder and a decoder, is an unsupervised deep neural network that
learns how to efficiently compress input data into a meaningful representation and subsequently
reconstruct the original data from this compressed form (Chen & Guo, 2023). By connecting the encoder
and decoder, the autoencoder effectively captures important patterns and variations present in the data,
enabling comprehensive analysis and interpretation (Chen & Guo, 2023). In this study, an autoencoder
neural network (figure 3) was built to analyze the measured VWC time series at 20 cm depth for the 9
sites.

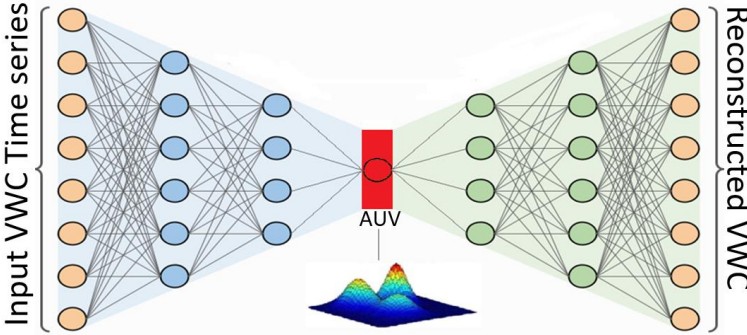

**Figure 3** Autoencoder deep neural network for volumetric water content dynamic analysis. In this
illustration, a densely connected autoencoder is utilized to compress the dynamic information of
Volumetric Water Content (VWC) into a singular value, AUV, highlighted in red. The process begins
with the encoder, depicted in blue, extracting the AUV from the measured volumetric water content
time series (left orange layer). Subsequently, the densely connected decoder, represented in green,
utilizes the AUV to reconstruct the VWC (orange layer at the right). Both the encoder and decoder,
characterized by dense connections, optimized the AUV value by minimizing the error between the
measured VWC and the reconstructed VWC.
The process was as follows. Firstly, an encoder neural network was created for each site. Its objective
was to take the VWC time series as input and gradually reduce its dimensionality through hidden layers
(Chen & Guo, 2023). The encoders' output was a single site-specific latent representation, called
Autoencoder Value (AUV), and captures essential features of the VWC dynamics (Chen & Guo, 2023).
Subsequently, a decoder neural network was developed to utilize the AUV value as reference to



reconstruct the original VWC time series data. The success of this reconstruction depends on the
training process, which aimed to optimize the AUV value by minimizing the error between the original
VWC time series and its reconstructed counterpart by minimizing the mean squared error (MSE) value
to less than 0.1.
After the optimization process, for each site one autoencoder value (AUV) was obtained. These AUV
were scaled and then used to build a combined model (Figure 4) as follows. The AUV were sorted into
three categories. Subsequently, one site from each category was selected. Finally, the data from the
three chosen sites, each representing one category, were used to train the combined AUC-DNN model.
The final combined model was thus equipped with 8 input parameters to predict the dynamic MP for a
specific location. These parameters consisted of the same 7 inputs employed in the DNN model (section
2.2), complemented by the AUV. The neural network structure, as detailed in section 2.2, remained
unchanged, employing the same optimization techniques.

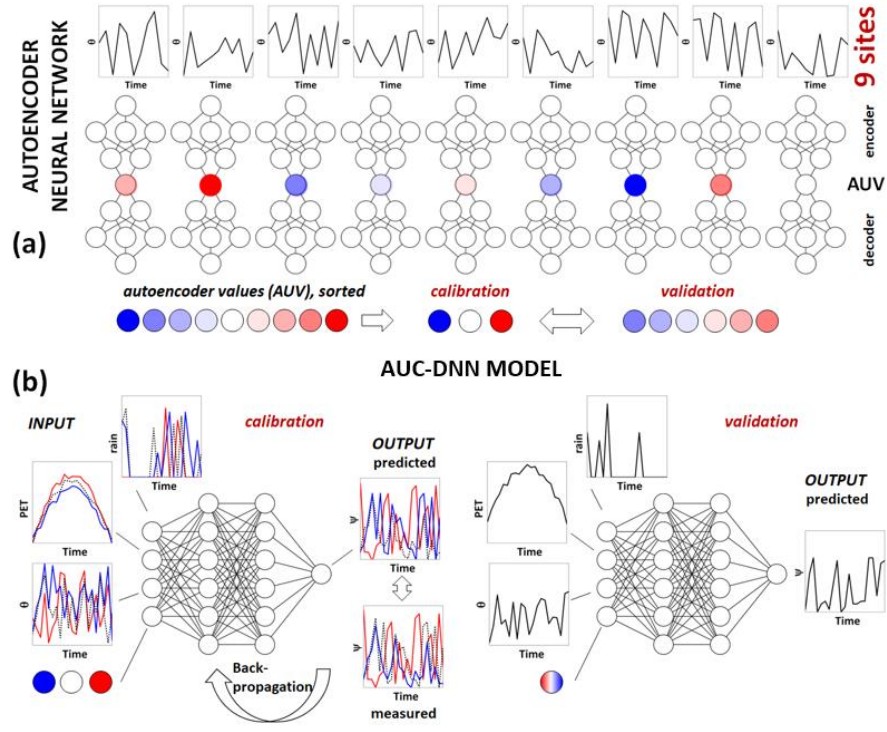




**Figure 4** Application of two different types of deep neural network for the prediction of matric potential
ψ. In this conceptual example, the water moisture dynamics of nine sites is considered. **(a)** The
autoencoder neural network captures the characteristic features of the soil water content (θ) dynamics,
assigning an autoencoder value (AUV) to each site. These values are sorted to AUV classes (one site
from each class was used for calibration, remaining sites for validation). **(b)** The combined AUC-DNN
model is built using the calibration sites with rainfall, potential evapotranspiration (PET), water content,
and AUV as part of the 8-input parameters. The predicted matric potential (ψ) is compared to measured
values for backpropagation. The calibrated DNN is then used to predict ψ for the remaining sites.
Initially, 70% of the data from each of the training sites were randomly selected for the training dataset.
Subsequently, the remaining 30% of the data were set aside as the holdout dataset, serving as a
benchmark for assessing model performance. The developed AUC-DNN was then applied for the other
six sites (with the same input variables including AUV) to predict the entire datasets of those unseen
sites. The combined model has thus the strengths of both components—the DNN' ability to understand
dynamic MP patterns and the feature extraction capabilities of the autoencoder. This shift in the model's
strength extends it from being site-specific to encompassing multiple sites, enabling it to gain a broader
understanding of how the dynamic MP and AUV values relate.
2.4 Statistical evaluation
The evaluation of model performance is carried out by comparing the model predictions to the measured
data. While there is no universal consensus on a standardized evaluation procedure, it is widely
recognized that a multi-objective approach should be adopted e.g., (Boyle, et al., 2000; Willems, 2009).
In this study, a combination of four evaluations tools was adopted. First, a scatter plot of observations
against simulated values was utilized to visualize the degree of alignment with the identity line (often
referred to as the 1:1 line). This graphical approach allowed for a qualitative assessment of model
performance. A closer concentration of data points near the 1:1 line indicated higher agreement between
calculated and observed values. Moreover, this graphical method includes the 95 % confidence interval
area which help in scrutinizing the model's consistency across different prediction ranges and detecting
potential biases within the model's performance (Ritter & Muñoz-Carpena, 2013).The second criterion
evaluates the distribution of (signed) prediction errors (eq(2)). Ideally, the error distribution should be
centered around zero, following a normal distribution pattern around this point with low standard
deviation. Such a distribution indicates an unbiased model with errors that tend to balance out.



Deviations from this pattern may suggest model bias or other unexpected characteristics in the
prediction errors PE (Ouden, et al., 2012).
$\text{PE} = O_i - P_i$                                                                                                  (2)
with observed $O_i$ and predicted matric potential value $P_i$. The third evaluation metric was the root
means squared error (RMSE; eq (3a)). RMSE with a value of zero indicates perfect fit, while higher
RMSE value means worse model performance (Ritter & Muñoz-Carpena, 2013). The final criterion for
model evaluation involved the use of the dimensionless goodness-of-fit indicator (eq (3b)), known as
the (Nash & Sutcliffe, 1970) coefficient of efficiency (NSE). NSE, which ranges from negative infinity
to 1, serves as an indicator of model performance, with a value of 1 indicating a perfect fit, while a
negative NSE suggests that using the means of the observed values is a better representative for the data
than the evaluated model itself (Ritter & Muñoz-Carpena, 2013; Gupta & Kling, 2011). An NSE value
of ≥ 0.55 was established as threshold for a good performance (Jiang, et al., 2020) and an NSE value >
0.80 as criterion for an optimal model. The RMSE and NSE are defined by:
$RMSE = \sqrt{\frac{\sum(O_i - P_i)^2}{N}}$                                                              (3a)
$NSE = 1 - \frac{\sum(O_i - P_i)^2}{\sum(O_i - \bar{o})^2}$                                              (3b)
where $O_i$ represents the measured value, $P_i$ the simulation output, and $\bar{o}$ the mean of the observed values,
all within the context of a sample size N.

## 3. Results

Following the model discussion in section 2.2 and 2.3, we present first the results of the site-specific
tests of predicting matric potential dynamics with a deep neural network (water content, rainfall and
evapotranspiration as input data), before the role of autoencoder value is considered.

### 3.1 Deep neural network modeling without autoencoder

The site-specific DNN model was used to simulate the time series for all nine sites. In Figure 5, the
results are shown for the Stüsslingen site (size #8, clay loam, meadow). The model was trained on data





that had 1825 days of observations from January 2012 to January 2020. The data was split randomly
into two parts: 1) a calibration dataset that had 1277 days and 2) a holdout dataset that had 548 days.
The model was then validated on data from February 2018 to January 2023 (1379 days). A strong
agreement between the model and the observed data was discovered in both the training and validation
datasets (figure 5c) as reflected by the low RMSE value and the high NSE value (table 1). Furthermore,
it was noticed that the error distribution exhibited a predominantly normal pattern with minimal bias
towards higher observed values compared to the predicted values (figure 5d). These findings suggest
that the site-specific DNN-model was not only able to be generalized well to unseen data but also
demonstrated a reliable ability to predict MP.
The statistical evaluation (Table 1) reveals a consistent performance across both the training and
validation periods for the Stüsslingen site, offering compelling evidence that the model avoids
overfitting. Additionally, when it comes to predicting MP values, the 95 % confidence interval indicates
that the model can capture well the overall dynamics (Figure 5b). However, the model performance
exhibits higher deviations for values exceeding 400 cm and consistently underestimates values higher
than 600 cm (figure 5b), which could explain the mild positive skewness observed in the distribution
of prediction errors in figure 5d.

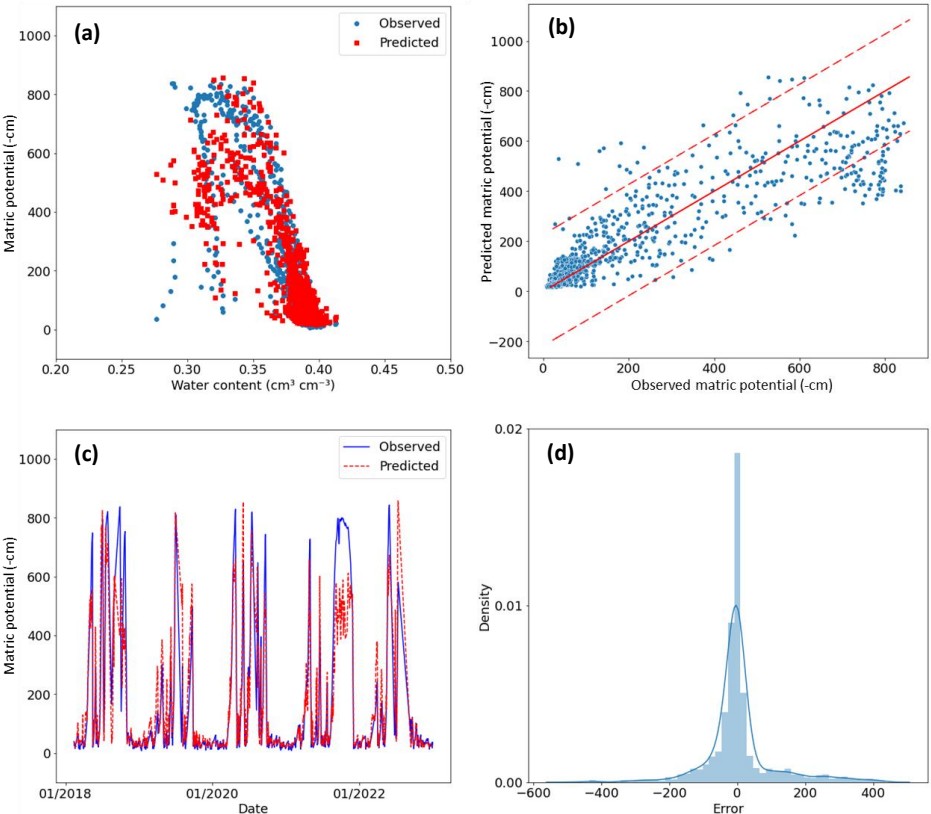


**Figure 5** Graphical evaluation of the performance of the site-specific deep neural network (DNN) for validation for the Stüsslingen site (site #8) for the validation period 2018 to 2022. **(a)** Comparison between the simulated and measured soil water characteristics curve. **(b)** Scatter plot comparing simulated and measured matric potential values, providing a visual representation of the level of conformity to the identity line. The two dashed lines represent the 95% confidence interval around the identity line, providing a visual assessment of the level of agreement. **(c)** Model validation presenting time series with the observed and predicted matric potential. **(d)** Analysis of the distribution of prediction errors (observed minus predicted values) with positively mild skewed distribution

Comparing the performance for the 'holdout' period (randomly chosen days between 2012 and 2019) of the nine site-specific DNN models, the NSE index is larger than 0.55 ('good') for all and larger than 0.80 ('optimal) for six sites. For all sites it was thus possible to build a DNN model with good model performance for the randomly chosen test days. However, for the validation period, only four showed optimal performance (NSE > 0.80). For two forest sites with an optimal performance for the holdout period (Dulliken, site #4, and Etziken, site #5), the NSE dropped from a range between 0.82 and 0.88 to a range between 0.73 and 0.75 (table 1). Obviously, the model captured the overall short term dynamics during training (randomly chosen days) but faced problems in the precise prediction of the





long validation period. An extended training period may be necessary to enhance the model's accuracy
for these specific sites. Three grassland sites (Bellach, site #2, Matzendorf, #6, and Hofstetten-Flüh, #5)
showed good but not optimal performance already during the holdout period. As discussed in the next
section, this may be related to large variations of the pressure values for similar water contents and the
corresponding large AUV. Notably, the lower performance observed in the holdout period for
Hofstetten-Flüh could be also linked to data limitations, as only 1200 days were used to train the model
for this specific site (compared to 1825 sites for the other sites).
**Table 1** Statistical assessment of calibration (1825 days, until year 2019/2020) and validation results
(years 2018/2019/2020 until years 2020/2021/2022) for nine sites. The holdout dataset was part of the
training period and includes 548 days (30 % of calibration).

| Location | AUV (-) | Training (holdout) | | Validation | |
|---|---|---|---|---|---|
| | | NSE (-) | RMSE (-cm) | NSE (-) | RMSE (-cm) |
| 1 Aetigkofen | 1.95 | 0.92 | 48 | 0.89 | 60 |
| 2 Bellach | 7.00 | 0.70 | 98 | 0.62 | 125 |
| 3 Breitenbach[a, b] | 3.56 | 0.86 | 82 | 0.83 | 96 |
| 4 Dulliken[a] | 2.19 | 0.82 | 55 | 0.73 | 103 |
| 5 Etziken[a] | 1.90 | 0.88 | 56 | 0.75 | 70 |
| 6 Hofstetten-Flüh[b] | 5.59 | 0.76 | 90 | 0.63 | 123 |
| 7 Matzendorf | 6.39 | 0.76 | 83 | 0.59 | 133 |
| 8 Stüsslingen | 4.49 | 0.80 | 71 | 0.80 | 98 |
| 9 Zunzgen | 6.44 | 0.87 | 62 | 0.83 | 73 |

[a] forest sites.
[b] Sites with limited available data. For those sites, only 1200 days were used for training; Within this training period, a subset
of 360 randomly selected days was designated as a holdout dataset; the validation period for those specific sites was from
2018/2019 to 2022.
3.2 Autoencoder DNN
The Autoencoder values (AUV) deduced from the time series analysis of the volumetric water content
for the period 2012-2022 can be classified in three main groups (figure 6). Soil water characteristics
curves (SWC) with low water content at saturated conditions and a small variation of water content for
similar potential values are assigned to 'type 1', contrasting 'type 2' with large water content values
and variations. These types of SWC are related to small ('type 1') and high ('type 2') autoencoder
values (AUV). Sites with AUV between these two classes, are denoted in the following as 'transitional'
type. As shown in Table 1, the AUV of forest soils are small (mainly 'type 1') with large NSE values.
In contrast to the forest soils, there are grassland sites with high AUV ('type 2') but small NSE.



Probably, the high variations of the SWC curve for 'type 2' require longer training periods to capture
the high variations in the pressure-saturation relationship.

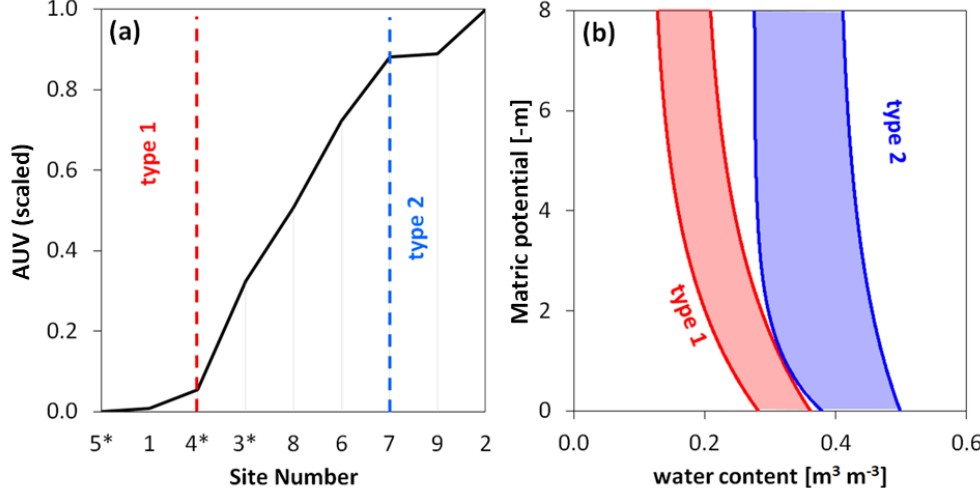

**Figure 6** Autoencoder value (AUV) and its relation to the soil water characteristics curve (SWC). **(a)**
The AUV of the nine sites with three sites of small (type 1) and three sites of high (type 2) AUV. **(b)**
The type 1 of the SWC has small water contents close to saturation and a narrower range of water
contents for similar water contents compared to type 2 with high water content values and variations.
Type 1 shows the data range of Aetigkofen (site #1) and Type 2 for Bellach site (#2). The site numbers
are chosen in alphabetic order and as shown in Figure 1 (Aetigkofen (1), Bellach (2), Breitenbach (3),
Dulliken (4), Etziken (5), Hofstetten-Flüh (6), Matzendorf (7), Stüsslingen (8), Zunzgen (9); sites with
forest are marked with *).

### 3.3 Deep neural network using the autoencoder value (AUC-DNN)

As mentioned in the previous section, the nine sites could be grouped into three main types according
to the scaled autoencoder value (AUV). Consequently, it was assumed that the creation of a DNN
model, which incorporates AUV in conjunction with the previously built site-specific neural network,
could enable predictions for unseen sites. Ideally, the model should be trained with a balanced dataset,
including one site from the 'type 1' category, one site from the 'type 2' category, and a few sites from
the 'transitional' category to capture the full transition between the 'type 1' and 'type 2'. However, due
to the data limitation, the model was trained for only three sites representing the three types (Etziken,
site #5, for 'type 1'; Bellach, #2, for 'type 2'; Stüsslingen, #8, for the 'transitional type') and was then
used to predict the six unseen sites. The impact of the small training set (only one site for transitional
type) was clear in the model results, which exhibited some instability, changing from one run to another
as the model was not able to assume the same transitional function between sites consistently. Therefore,



the model was run 20 times, then the average result for theses runs was taken as a representative
outcome. The application of the new DNN model with AUV to predict the dynamic of matric potential
is shown in Figure 7 for Breitenbach (site #3, loam, forest) as unseen site. The model was found to fell
slightly behind the previously designed DNN model, but still can predict the dynamic in a good way.
Notably, the NSE value for this model for Breitenbach site was 0.71 over the entire period from 2012
to 2022 (Table 2).
**Table 2** AUC-DNN Model performance for the period 2012-2022. Three training sites were used to
build the AUC-DNN model that was then applied for the other six sites. The sites are listed according
to the corresponding autoencoder value (AUV). The asterisks mark the sites with forest; The AUV
was scaled from 1.9 to 7.0 to simplify input. Alternatively, scaled values ranging from 0 to 1 could
also be utilized.

| Location | AUV | AUV (type) | used as | NSE (-) | RMSE (cm) |
|---|---|---|---|---|---|
| 5 Etziken* | 1.90 | Type 1 | Training site | 0.82 | 70 |
| 1 Aetigkofen | 1.95 | Type 1 | Validating site | 0.76 | 88 |
| 4 Dulliken* | 2.19 | Type 1 | Validating site | 0.65 | 100 |
| 3 Breitenbach* | 3.56 | Transitional | Training site | 0.71 | 73 |
| 8 Stüsslingen | 4.49 | Transitional | Validating site | 0.85 | 116 |
| 6 Hofstetten-Flüh | 5.59 | Transitional | Validating site | 0.60 | 113 |
| 7 Matzendorf | 6.39 | Type 2 | Validating site | 0.58 | 123 |
| 9 Zunzgen | 6.44 | Type 2 | Validating site | 0.69 | 104 |
| 2 Bellach | 7.00 | Type 2 | Training site | 0.71 | 104 |


It was noticed that the error distribution exhibited a predominantly normal pattern with a bias towards
higher observed values compared to the predicted values (figure 7d). The analysis indicates the model's
proficiency in forecasting dynamic trends rather than precise values (figure 7c). The results align with
the anticipated scenario as the AUV for Breitenbach (3.56) was relatively close the Stüsslingen AUV
value (4.49). Therefore, the underestimation detected in Stüsslingen for the site-specific DNN (figure
5b) is expected to exist in Breitenbach as well. The average model performance for all sites is presented
in Table 2. The NSE values was > 0.55 for the 6 unseen sites (validating sites) and provided strong
evidence that the model can be relied upon for the dynamic MP predictions.





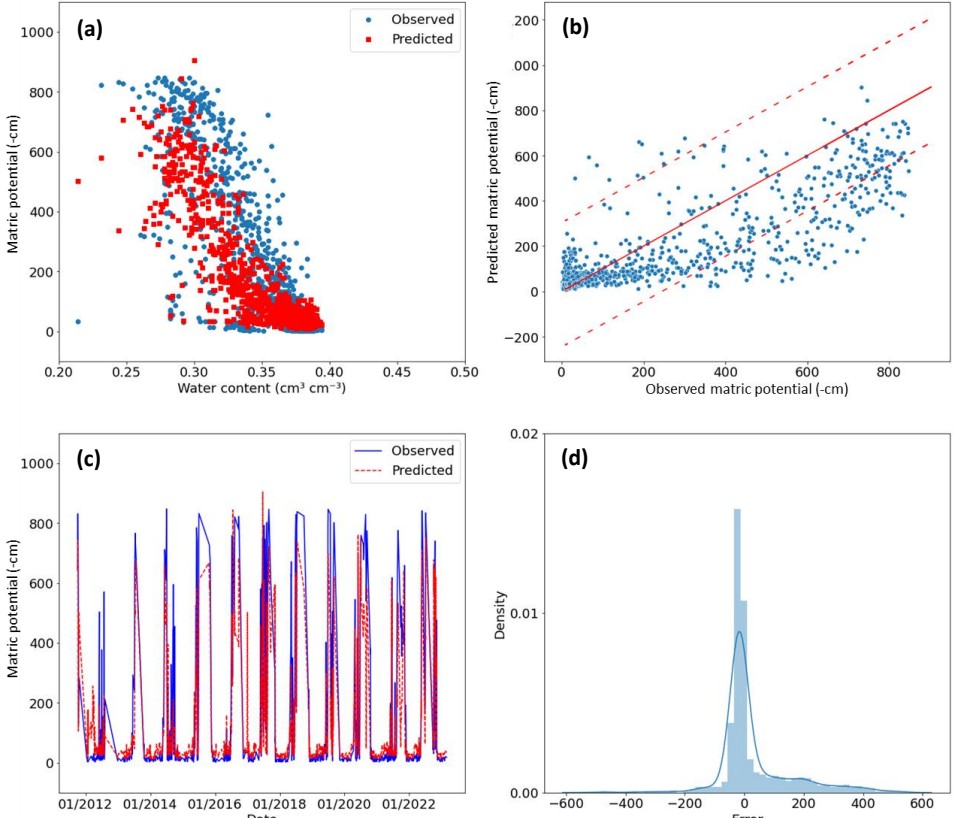

**Figure 7** Evaluation of the Deep Neural Network with Autoencoder (AUC-DNN) model performance at the Breitenbach site for the period 2012-2022. **(a)** Comparison between the expected Soil Water characteristics curve (SWC) and the observed SWC. **(b)** Scatter plot that compares observed data points with their corresponding simulated values, providing a visual representation of the level of conformity to the identity line. The two dashed lines represent the 95% confidence interval around the identity line, providing a visual assessment of the level of agreement. **(c)** Time series comparison showing the observed and predicted matric potential for the entire period. **(d)** Analysis of the distribution of prediction errors (observed minus modelled value) using positively mild skewed distribution.

The NSE values for the unseen sites (validating sites) varied from 0.58 to 0.76, indicating a spectrum

of model performance, ranging from acceptable to good. The low NSE values observed for Matzendorf

(site #7) suggest that the model's utility is more suited for capturing overall trends and dynamics rather

than precise values. This evaluation was further supported by examining a scatter plot (Figure 8) that

compares the observed data points with their corresponding simulated values for the sites scored the

lowest and the highest NSE, Matzendorf (site #7) and Aetigkofen (site #1). The plot revealed a wider

95% confidence interval for Matzendorf (figure 8a) in comparison to Aetigkofen (figure 8b), indicating

that the lower the NSE value is, the more challenging it became for the model to predict the exact MP



values. However, the model performance indicated the ability of the AUC-DNN model to predict
dynamic MP without the necessity of site-specific training data, marking a transition from the DNN
site-specific nature to a more versatile multi-site model.

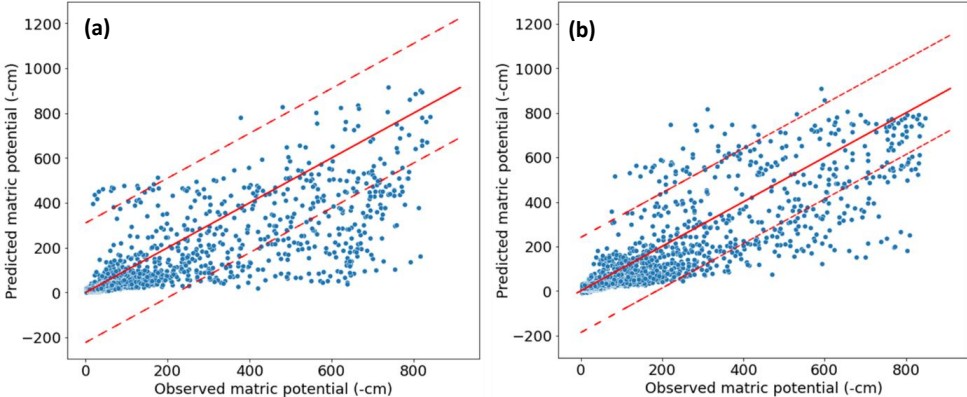


**Figure 8** Comparison between observed data points and their corresponding simulated values for two
sites with lowest and highest efficiency coefficient NSE. **(a)** Matzendorf (site #7) with NSE of 0.58. **(b)**
Aetigkofen (site #1) with NSE of 0.76. The solid lines mark the 1:1 correspondence, the dashed lines
the 95% confidence interval.
## 4. Discussion
Based on the analysis of the simulation results presented in section three, it can be asserted that the
model was successfully built. However, as discussed in the next subsection, the model is expected to
have certain drawbacks due to the limited number of available sites. In the other subsections, the
relationship between the autoencoder value and soil properties and its application for satellite data will
be discussed.
### 4.1 Limits of the deep neural network with autoencoder value (AUV-CNN)
First, the model's statistical evaluation revealed that the matric potential (MP) at a depth of 20 cm could
be simulated with acceptable precision. However, a high variability in the evaluation is indicated by the
NSE values for the unseen sites. This variance is attributed to the model's limited generalization
capacity, as it was trained on just three sites. Furthermore, the model was not able to catch the whole
dynamic for the training sites due to the limited length of available data. For example, Bellach (site #2),
a training site that has a high AUV value, had NSE value of 0.71 for the training period (table 2), which



indicates that the model was able to catch the general trend for this site, but still can't predict the exact
value of the MP. The effect of this result was obvious on the sites that are closed to AUV 'type two'
category (e.g., Hofstetten-Flüh and Matzendorf, sites #6 and #7, with NSE of 0.60 and 0.58,
respectively).
The stability of the AUC-DNN model was insufficient, as the model showed different prediction quality
upon running the model repeatedly for the same training sites (figure 9). This variability in the outcomes
indicates that the model can find different MP dynamics scenarios inside the training data. Therefore,
it is recommended to train the model for more than one site in the same AUV type.

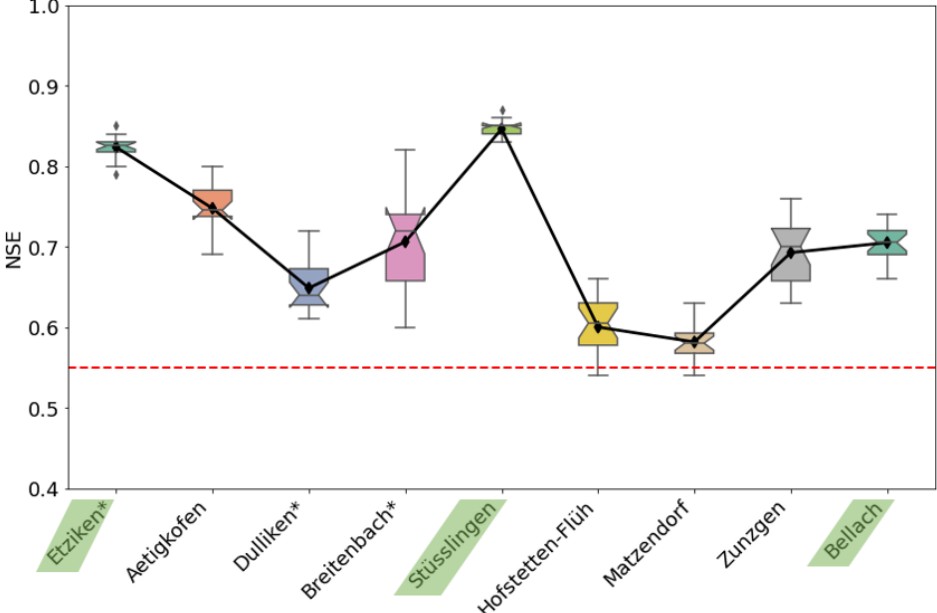


**Figure 9** Variation of prediction results for 20 Runs for the AUC-DNN model quantified with the
efficiency coefficient NSE. The highest variation was with the unseen sites in the transitional and type
2 categories. Each box represents the interquartile range, with the line inside denoting the median. The
black diamond markers connect the mean values for each station, providing insight into the central
tendency of the data. Notches on the boxplots offer a visual indication of the uncertainty around the
median. The red dashed line represents the defined threshold for the NSE, set at 0.55 ; sites with forest
are marked with *; training sites are highlighted in green.
Especially for the 'transitional type', choosing a site in the beginning, in between, and in the end of the
category would stabilize the modeling results. However, in this study, there was no possibility to
provide the model with extra data to solve the prediction instability. Therefore, a solution was
implemented by 1) closely monitoring the model manually to ensure it captures the dynamic from all



three sites. This involved training the model with nearly identical time periods for each site and visually
confirming comprehensive coverage of the cloud of points for the retention curve of each site, avoiding
concentration on specific patterns during training. The process also includes 2) running the model for
20 times, then averaging the results. Additionally, the statistical evaluation plots as shown in Figure 8,
were used to detect instances with very low or very high MP prediction values.
4.2 Relationship between AUV and physical soil properties
As discussed in section 3.2, the autoencoder value (AUV) is low for soil water characteristics curves
with low saturated water content and low variations of water content for a certain matric potential value
(and high AUV for large values and variations of water content). To define a more quantitative
relationship between SWC and AUV, the SWC data were characterized as follows: the time average of
volumetric water content (VWC) and SWP were calculated for 15 days for the period 2015 to 2022.
The envelope of these data was then calculated by fitting a minimum and maximum pressure saturation
relationship including the averaged data (see Figure 10a).

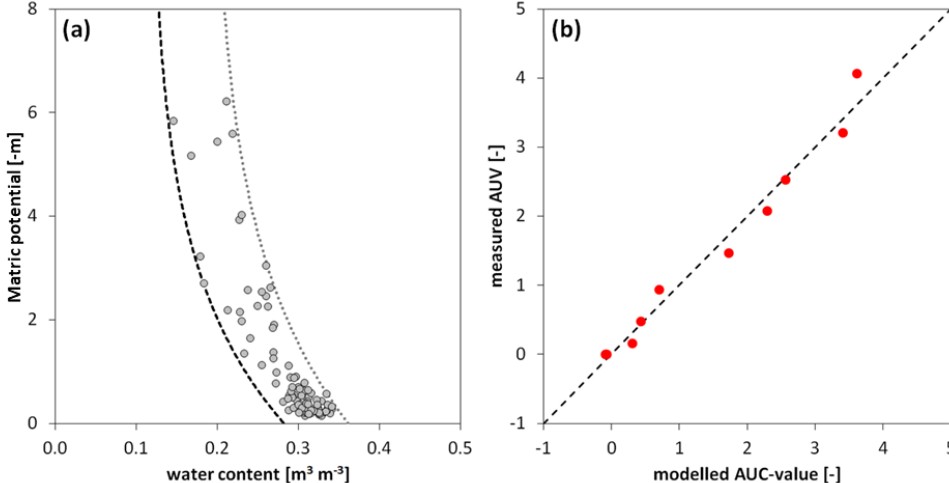

**Figure 10** Relationship between autoencoder value (AUV) and soil water characteristics curve (SWC).
**(a)** 15-days average of SWC data for Aetigkofen (symbols; site #1). The two lines are exponential
functions building the envelope of the SWC curve. **(b)** Linear model for the nine sites linking the
parameters of the exponential model with the 'measured' AUV (deduced from measured water content
data).
The two boundary lines were characterized by a 'saturated' and 'residual' water content and a shape
parameter defining an exponential decrease of water content with increasing absolute matric potential





values. The SWC of each site can thus be described by six parameters (three parameters per boundary
line). As shown in Figure 10b, a linear model expressing the AUV as function of these six parameters
can be built. It was not possible to reproduce the AUV as linear model of soil texture and average water
content, indicating that the soil moisture dynamics represented by AUV is not only dependent of static
soil textural attributes but seasonal structural features as well.
### 4.3 Application for satellite data
The AUC-DNN model was used to analyze satellite-based volumetric water content (VWC) satellite
data, including SMAP L4 and L3, SMOS products, and Sentinel data. Subsequently, a comparison was
carried out for the AUV for both site-specific measurements and earth observation (EO) measurements
for the same region. The initial findings highlighted a disparity between the dynamics captured by EO
products and the actual dynamics. Therefore, if the objective is to establish a robust system capable of
detecting changes in water retention dynamics on a regional scale, it is considered necessary to enhance
the calibration of EO in Europe. Only with EO-data that can reproduce the essential of the soil moisture
dynamics as manifested in the AUV, the matric potential dynamics can be deduced from EO-data. For
future EO-data with improved capacity to capture regional soil moisture dynamics, the concept
presented in this study (AUC-DNN) could be used to predict matric potential dynamics at global scale
(see Appendix C).
## 5. Summary and conclusions
The soil water potential (SWP) determines water flow direction, water ability for plants, and mechanical
stability. Because it cannot be measured directly by remote sensing techniques at larger scales, it is
often deduced from water content information, assuming an unambiguous relationship between water
content and SWP. However, this relationship under dynamic field conditions is highly ambiguous due
to hysteresis, dynamic effects, and soil structural changes that cannot be modeled with a physically-
based model. To enable prediction of SWP from soil water content, we apply a deep neural network
(DNN) with an autoencoder to define unique features of the soil moisture dynamics. By inserting the
autoencoder value (AUV) together with climatic data and water content measured at nine sites in the



region of Solothurn (Switzerland) in a deep neural network (AUC-DNN), the soil water potential could
be predicted. The main findings of the study can be summarized as follows:
• The SWC of the nine sites can be classified in three types based on the width of pressure-

saturation relationship and the water content close to saturation

• These SWC-types are manifested in different autoencoder values (AUV)
• The AUV is not a simple function of average water content or soil texture but includes structural

effects as well

• The AUC-DNN model could predict successfully the SWP dynamics of sites without site-

specific training

The autoencoder value (AUV) is thus a new descriptor of the complex soil moisture dynamics that
cannot be captured with physically based models. Future satellite generation may be sensitive enough
to measure the AUV from remote sensing water content data. The approach presented in this paper will
then enable the prediction of the soil matric potential at the global scale using remote sensing water
content data.

## Appendix A: Data Quality Assurance and Trend Analysis


As a precaution for data quality, the Absolute Matric Potential (AMP) and volumetric water content
(VWC) data were scrutinized to identify potential errors the data. The process includes different steps
that were necessary to discover anomalies, checking the integrity of the data, and detecting systematic
changes with time.
**1- Flagging Abrupt Changes in VWC and MP:**
**VWC Flagging and removing:**
• Differences between consecutive (daily) time steps in the water content time series were

calculated.

• Instances with daily differences exceeding 0.1 $cm^3/cm^3$ were flagged and denoted as sudden

decreases or increases in VWC.





- Instances with VWC below 0.1 cm³/cm³ or exceeding 0.7 cm³/cm³ were identified and removed from the dataset. These extreme values were considered as measurement anomalies or outliers affecting the overall dataset's reliability.

- Instances with AMP<1 cm was removed from the data to overcome limitations in the used method. The water potential can change without modifying the volumetric water content after this limit, which could make the results of the model not accurate enough.

- The differences between consecutive time steps in AMP -time series was calculated; instances with daily differences exceeding 500 cm were flagged and called sudden decreases or increases in AMP (figure A1).

- The threshold AMP-value of 850 cm was employed in a specific step, where instances with AMP exceeding 850 cm were removed from the dataset, addressing the physical properties of water as it starts to boil in the tensiometers under pressure after this limit.

- Periods of concurrent decrease in AMP (indicator for wetting) and decrease in VWC (drying) were flagged (figure A1).

- Periods with matric potential values remaining constant over a three-day rolling window were flagged (figure A1).

**2- Utilizing Index Windows for Data Manipulation and Data Removal**

To address flagged instances mentioned before, a systematic approach is employed. For each flagged instance, three additional indices are generated around it to construct an index window, spanning one day before (index_1), the flagged instance itself (index_0), and two days after (index_2 and index_3). This four-day index window was eliminated from the dataset (figure A1). The decision to eliminate this window was informed by a visual assessment of measurements as it was noticed that when a measurement error occurs, the accuracy of the preceding day is affected. Furthermore, it was assumed that the device requires two subsequent days to restore normal measurement precision. This process contributes to a refined dataset, providing a more accurate representation of the underlying trends in AMP and VWC.



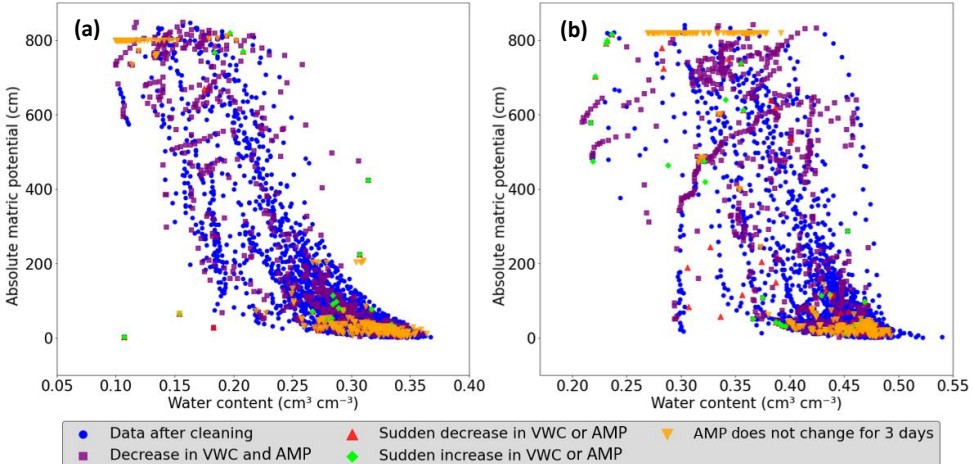

**Figure A1** Comparison of data before and after cleaning procedure: the blue circles depict the remaining data after applying the cleaning criteria. Each distinct marker represents eliminated points, each corresponding to a specific criterion (e.g., the square purple marker for simultaneous decrease in volumetric water content (VWC) and the absolute matric potential (AMP), the red upward-pointing triangle is the marker for sudden decreases, the lime diamond for sudden increases, and the orange downward-pointing triangle marks periods of unchanged AMP). This provides insights into the reasons for data removal and illustrates the profound impact of the data cleaning process in retaining high-quality data points. In **(a)** the cleaning process for sandy clay loam site in Aetigkofen (site #1) is shown, in **(b)** the cleaning process for the Matzendorf site (site #9, clay loam soil).

## Appendix B: Running the model with Logarithmic MP value.

The AUC-DNN showed a good performance in predicting the dynamic MP for the different 6 unseen

sites. However, it was clear that the model prioritizes tends to focus on capturing significant changes in

values rather than accurately representing the values themselves. This tendency is attributed to the

substantial difference between the highest and lowest absolute values (approximately 850 cm), leading

the model to emphasize major fluctuations while neglecting minor ones. To address this issue and

enhance the model's precision in capturing the exact AMP, a suggestion has been made to train the

model for the same three sites but with the logarithmic value for the AMP. This modification aims to

strike a better balance, ensuring that both major and minor changes are effectively captured while

maintaining accuracy in representing the specific values of MP.

To qualitatively assess the model training performance under the logarithmic scale, a scatter plot (Figure

B1) was generated, comparing observations against simulated values for the second training site

(Stüsslingen). The reason for choosing a training site was to understand how the model captures the



dynamics when trained with logarithmic matric potential. The results suggest that using logarithmic
scale, the model prioritized the prediction of the exact absolute value of matric potential (AMP), which
makes the model to optimize predictions for the absolute values between 0 to 200 cm. This approach is
giving the same importance to small and large changes in the AMP, which causes that the model
assigned a higher weight to small changes according to their higher frequency, while neglecting less
frequently occurring major dynamic shifts. Consequently, the model's accuracy went down beyond 200
cm (figure B1a) when compared to the model trained on non-logarithmic AMP-values (figure B1b). To
maintain a balanced consideration of changes, logarithmic MP was avoided in the main part of the
paper.

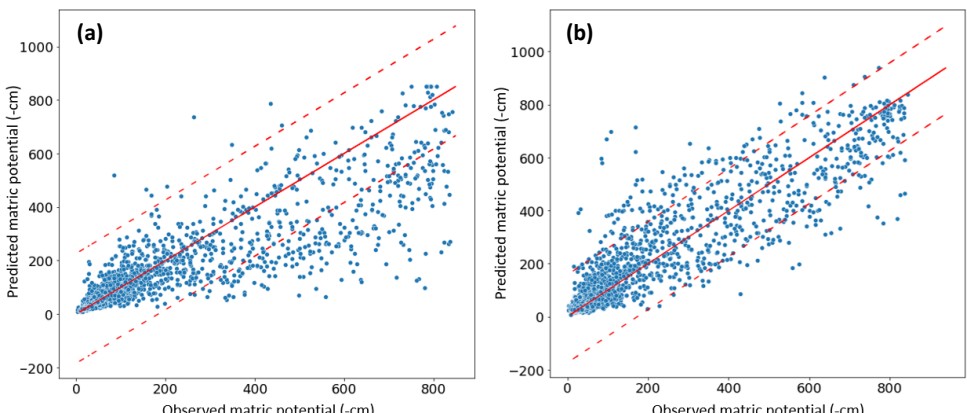

**Figure B1** Visual comparison of model performance, comparing the observed and simulated values for
the Stüsslingen training site. **(a)** the model trained with logarithmically scaled AMP-values, while in
**(b)** The model trained with absolute linear matric potential (AMP) values. The solid line denotes the
1:1 correspondence, and dashed lines represent the 95% confidence interval.
## Appendix C: SMAP data and Autoencoder for global scale analysis
SMAP (Soil Moisture Active Passive) is a NASA satellite mission that was established to help in
improving weather forecasts and global drought monitoring. SMAP data products are available at
different levels of processing, from Level 1 (L1; instrument measurements) to Level 4 (L4; model-
derived value-added products). For this study, SMAP L3 and SMAP L4 products for measuring
moisture content were used. The main difference between the two products is that SMAP L3 depends
on the passive radiometer measurements, while SMAP L4 products are derived from a data assimilation



system that combines the L-band brightness temperature observations from SMAP with a land surface
model and meteorological forcing data (Reichle, et al., 2019). SMAP L3 products for moisture content
are primarily affected by vegetation and surface roughness, allowing them to capture surface soil
moisture variations. In contrast, the incorporation of land surface models in SMAP L4 products reduces
its sensitivity to vegetation covers and surface roughness, making the products more representative of
the profile soil moisture conditions (Reichle, et al., 2019; Sadri, Wood, & Pan, 2018).
The autoencoder's encoded representations offer a unique opportunity to compare the spatial patterns
inherent in "point measurement" with remote sensing data such as SMAP L3 and SMAP L4 data. The
autoencoder method could illuminate how these diverse data streams align or diverge, providing crucial
insights into the compatibility and complementarity of ground and satellite measurements. The process
was applied for the data between the years 2015 to 2022. All the data (SMAP L4, SMAP L3, and on-
site measurements) were given to the autoencoder neural network together. Subsequently, the resulting
autoencoder values were scaled. Finally, a comparison was made to show if the satellite measurements
and the on-site measurements have the same measured dynamics.
The autoencoder analysis of SMAP L3 (figure C1) indicates that satellite measurements struggle to
capture the dynamic change of the water content, as all locations yield approximately the same
Autoencoder Value (AUV). In contrast, the SMAP L4 product (figure C1) exhibits fluctuations in AUV
results. For instance, Stüsslingen and Matzendorf align closely with on-site measurements in terms of
AUVs. However, for Hofstetten-Flüh, the SMAP L4 product indicates a very small AUV, suggesting
an expected dynamic in line with a type 1 soil water retention curve (figure 6b). In contrast, on-site
measurements indicate a higher AUV for Hofstetten-Flüh, suggesting a closer association with a type
2 soil water retention curve. These findings underscore the imperative for developing a new
methodology to calibrate satellite data in the Switzerland area. The prevalent uniformity in SMAP L3
results and the notable disparities between on-site measurements and satellite data across various
products highlight the need for a more refined approach to ensure accurate and reliable dynamic soil
moisture assessments.





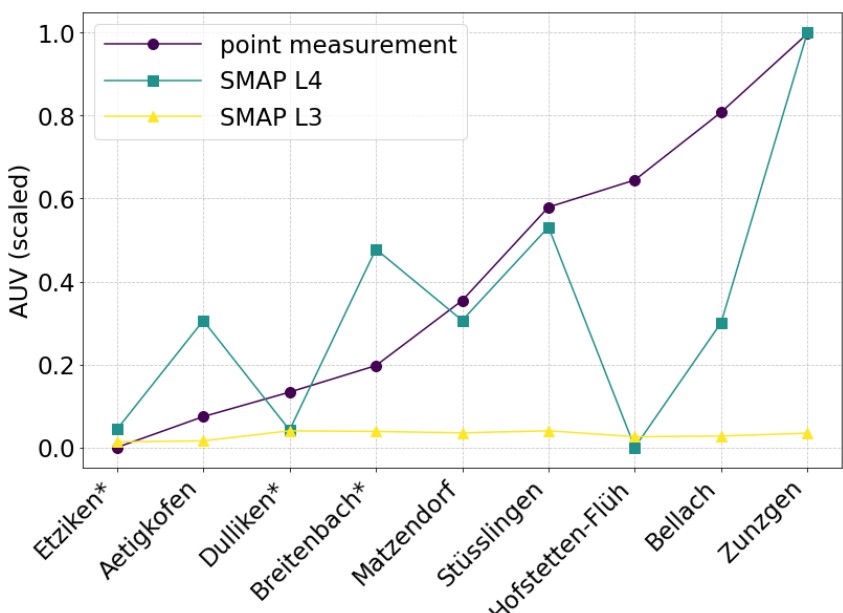

**Figure C1** Comparative analysis of Autoencoder Neural Network results for SMAP L3 and SMAP L4 satellite data, alongside with profile measurements. The fluctuating AUV values indicate varying degrees of alignment with on-site measurements across different locations. Sites with forest are marked with *.

## Code and data availability

The related input data for the AUC-DNN model and Python code are openly accessible under

https://doi.org/10.5281/zenodo.10600669 and https://doi.org/10.5281/zenodo.10602397 respectively.

The input for the autoencoder and its python codes are openly accessible under

https://www.doi.org/10.5281/zenodo.10605108

## Author contributions

NA, AC, and PL designed the research. NA and PL performed the research. NA and MR analyzed the

soil moisture time series. SM was responsible for the soil moisture network. NA wrote the codes and

built the model. NA and PL wrote the manuscript with substantial input from all co-authors.

## Competing interests

The contact author has declared that none of the authors has any competing interests.



## Financial support

"This research is part of the project Artificial Intelligence for Soil Health. Funded by the European Union. Views and opinions expressed are however those of the author(s) only and do not necessarily reflect those of the European Union or of the Research Executive Agency (REA). Neither the European Union nor the granting authority can be held responsible for them."

## Acknowledgements

NA acknowledges the utilization of ChatGPT to enhance coherence within certain sections of the manuscript.



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
