# Peer review of "Prediction of Hysteretic Matric Potential Dynamics Using Artificial 1 Intelligence: Application of Autoencoder Neural Networks 2 3 Nedal Agel1\*, Lea Reusser1,2,3, Stephan Margreth2, Andrea Carminati1, and Peter Lehmann1 4 5 1<"

_EGUsphere, 2024_

## Author Response (AR1)

**REVIEW 1 (our reply in blue, the relevant changes made in the manuscript in red)**

**Summary**

Aqel et al. present a neural network (NN) based approach to predict matric potential from soil water
content observations. Using an autoencoder, they extract the most relevant features of the soil water
retention dynamics. They input their results into a deep neural network (DNN), which increases
the transferability of the DNN.

**Assessment**

The approach presented in this paper is convincing. The manuscript is well-written. Prediction of
hysteresis in soil water retention is of interest to the soil hydrology and soil physics community.

I don't have major comments. Thus, I recommend accepting the manuscript after minor revisions.
I have some minor comments below.

We thank the reviewer for the positive feed-back and the specific comments (they are addressed
below).

**Nash-Sutcliffe efficiency**

The Nash-Sutcliffe coefficient tends to emphasise maxima in a time series, which might bias the
results. An additional interesting metric would be the Kling-Gupta efficiency (Knoben et al., 2019;
doi: 10.5194/hess-23-4323-2019).

Thank you for the input. We computed the Kling-Gupta Efficiency (KGE) for the nine sites (see Table
1 on the next page).

KGE was developed to address some limitations of the Nash-Sutcliffe Efficiency (NSE) by
incorporating three components: correlation, bias, and variability (Liu, 2020;
https://doi.org/10.1016/j.jhydrol.2020.125488 ). The KGE provides a more comprehensive assessment
of model performance by balancing these aspects. The value of KGE ranges from negative infinity to
1, with a value of 1 indicating perfect agreement between observed and modeled data.

In a study by Gupta et al. (2009; https://doi.org/10.1016/j.jhydrol.2009.08.003), a KGE of 0.6 was
considered acceptable for streamflow simulations. In our analysis, we follow Towner et al. (2019,
https://hess.copernicus.org/articles/23/3057/2019/) that used KGE > 0.75 as threshold for "good" model performance. This threshold suggests that the model accurately captures the dynamics of the observed data, including the mean, variability, and correlation structure.

**KGE values for deep neural network modeling without autoencoder**

After running the deep neural network model (section 3.1), the KGE values were as shown in Table 1.

Only two sites, Matzendorf (site #7) and Etziken (site #5), from the nine sites had a KGE value of less than 0.75 in the validation (KGE > 0.75 for training). These two sites were mentioned in section 3.1 as sites needing more training data, which follows the expected scenario by NSE.

In conclusion, the KGE-analysis defines good model performance for seven out of nine sites (four sites according to NSE-criterion of NSE ≥ 0.80).

*Table 1: Statistical assessment of calibration (1825 days, until year 2019/2020) and validation results (years 2018/2019/2020*

*until years 2020/2021/2022) for nine sites. The holdout dataset was part of the training period and includes 548 days (30 %*

*of calibration).*

| Location | AUV (-) | Training (holdout) | | | Validation | | |
|---|---|---|---|---|---|---|---|
| | | NSE (-) | RMSE (cm) | KGE (-) | NSE (-) | RMSE (cm) | KGE (-) |
| 1 Aetigkofen | 1.95 | 0.92 | 48 | 0.91 | 0.89 | 60 | 0.87 |
| 2 Bellach | 7.00 | 0.70 | 98 | 0.84 | 0.62 | 125 | 0.77 |
| 3 Breitenbach[a, b] | 3.56 | 0.86 | 82 | 0.78 | 0.83 | 96 | 0.84 |
| 4 Dulliken[a] | 2.19 | 0.82 | 55 | 0.86 | 0.73 | 103 | 0.76 |
| 5 Etziken[a] | 1.90 | 0.88 | 56 | 0.90 | 0.75 | 70 | 0.65 |
| 6 Hofstetten -Flüh[b] | 5.59 | 0.76 | 90 | 0.79 | 0.63 | 123 | 0.81 |
| 7 Matzendorf | 6.39 | 0.76 | 83 | 0.79 | 0.59 | 133 | 0.63 |
| 8 Stüsslingen | 4.49 | 0.80 | 71 | 0.84 | 0.80 | 98 | 0.85 |
| 9 Zunzgen | 6.44 | 0.87 | 62 | 0.82 | 0.83 | 73 | 0.77 |

[a] forest sites.

[b] Sites with limited available data. For those sites, only 1200 days were used for training; Within this training period, a subset of 360 randomly selected days was designated as a holdout dataset; the validation period for those specific sites was from

2018/2019 to 2022.

**KGE values for deep neural network using the autoencoder value (AUC-DNN)**

The results of the Kling-Gupta Efficiency (KGE) for the Deep Neural Network Autoencoder (AUC-

DNN) model in section 3.3 show that three out of the six validation sites have a KGE value less than

0.75 (Table 2, next page). The two sites Hofstetten-Flüh (site #6) and Matzendorf (site #7) have the lowest NSE values, indicating that the model captures the general dynamics of these site rather than the exact values. This is consistent with our conclusion in section 3.3. The third site, Breitenbach (site #3), was identified in section 3.3 as a site where underestimation is expected (see Figure 7 in manuscript), which explains why its KGE value is below the threshold of 0.75.

*Table 2: AUC-DNN Model performance for the period 2012-2022. Three training sites were used to build the AUC-DNN model that was then applied for the other six sites. The sites are listed according to the corresponding autoencoder value (AUV). The asterisks mark the sites with forest; The AUV was scaled from 1.9 to 7.0 to simplify input. Alternatively, scaled values ranging from 0 to 1 could also be utilized.*

| Location | AUV | AUV (type) | used as | NSE (-) | RMSE (cm) | KGE (-) |
|---|---|---|---|---|---|---|
| 5 Etziken* | 1.90 | Type 1 | Training site | 0.82 | 70 | 0.81 |
| 1 Aetigkofen | 1.95 | Type 1 | Validating | 0.76 | 88 | 0.76 |
| 4 Dulliken* | 2.19 | Type 1 | Validating | 0.65 | 100 | 0.77 |
| 3 Breitenbach* | 3.56 | Transitional | Validating | 0.71 | 73 | 0.68 |
| 8 Stüsslingen | 4.49 | Transitional | Training site | 0.85 | 116 | 0.91 |
| 6 Hofstetten-Flüh | 5.59 | Transitional | Validating | 0.60 | 113 | 0.72 |
| 7 Matzendorf | 6.39 | Type 2 | Validating | 0.58 | 123 | 0.56 |
| 9 Zunzgen | 6.44 | Type 2 | Validating | 0.69 | 104 | 0.81 |
| 2 Bellach | 7.00 | Type 2 | Training site | 0.71 | 104 | 0.80 |

Similar to the analysis without autoencoder discussed in the previous paragraph, the KGE-analysis defines good model performance for more sites than according to the NSE. Accordingly, the NSE-thresholds for good model performance are more challenging compared to KGE and we focus on NSE (and not KGE) in the paper and no relevant changes were made in the manuscript related to this comment.

**Local nature of the model**

The results obtained are from sites that share similar climate and topography. I wonder if this workflow would work as well in different regions of the world, or if further adjustments must be made.

To show what is needed to apply the workflow for other regions, we used the Deep Neural Network Autoencoder (AUV-DNN) model described in section 3.3 to predict the matric potential for a site called 'Wasen' in the hilly region around the Napf mountain in Switzerland. Compared to the canton of Solothurn (the sites presented in the paper), the Napf region has a different geology and is known to be colder in winter and having more rainfall.

The AUV value for Wasen was 2.54 and falls within the range of the training sites (1.90 to 7.00). Accordingly, we could expect that the soil moisture dynamics was similar to the sites in the canton of Solothurn. The model was thus able to predict the matric potential with high quality, as shown in the figure on the next page. The NSE for the model was 0.81, and the KGE was 0.89, indicating that the model performs well in predicting unseen sites in other locations.

However, Wasen is still in Switzerland, with soil texture similar to the training sites. It is expected that the model may not perform well for sites with AUV values outside the training range.

We commented on this in section 4.1 in the revised manuscript as shown below:

"For the set of sites analysed in this study, the model showed good generalization capacity and stability.

However, the nine sites were similar with respect to climate and geology and the range of soil textural classes (see Figure 1) was relatively narrow. In a future study, the AUC approach will be applied for sites differing in climate and soil textural classes. We expect that the model can predict the dynamic matric potential for a new site as long as the autoencoder value falls within the range of AUV of the training sites. To predict the soil moisture dynamics for soils with autoencoder values outside of the range of training data, the model must be re-built using additional training data."

[Figure]

*Figure 1. Evaluation of the Deep Neural Network with Autoencoder (AUC-DNN) model performance at the Wasen site for the*

*period 2019-2023. (a) Comparison between the expected Soil Water characteristics curve (SWC) and the observed SWC. (b)*

*Scatter plot that compares observed data points with their corresponding simulated values, providing a visual representation of the level of conformity to the identity line. The two dashed lines represent the 95% confidence interval around the identity line, providing a visual assessment of the level of agreement. (c) Time series comparison showing the observed and predicted matric potential for the entire period. (d) Analysis of the distribution of prediction errors (observed minus modelled value) using positively mild skewed distribution.*

**Physically-based modelling**

I partially agree that the reductionist mechanistic models might be unable to account for the full complexity inherent in the soil water retention process. Input-agnostic approaches such as neural networks surely have an advantage when it comes to predicting matric potential. However, physically-based modelling is also a tool for process understanding that could potentially help us disentangling the effects of all the interacting processes that control soil water retention. I know that there are efforts to make machine learning a tool for process understanding as well. Perhaps the authors could comment briefly on this and place their work in this discussion?

We agree with the reviewer that we need physically-based modelling for process understanding. Due to the complexity of the involved physical processes at the field scale (hysteresis, non-equilibrium, seasonal dynamics of soil structure), we don't have a yet a physical model to predict these processes. Machine learning could help to disentangle these effects, for example by classifying periods that are affected by structural changes and periods that are dominated by non-equilibrium effects. For the different periods, specific amendments in the description of the physical process and properties could be developed (i.e., the application of season-dependent and rate-dependent soil hydraulic properties).

Alternatively, physically-induced machine learning (PIML) should be applied in the future, to link the knowledge we have on the physical processes with the data-driven machine learning approaches. There are recent applications of PIML in hydrology: Degen et al. (2023; https://gmd.copernicus.org/articles/16/7375/2023/) replaced the complex numerical simulations of the Richards equation with a surrogate model using a set of physically-based basis functions; Bhasme et al. (2022; https://doi.org/10.1016/j.jhydrol.2022.128618) combined a set of simple physically-based mass balance equations with machine-learning to predict successfully evapotranspiration and streamflow from a catchment. A similar approach is possible for the problem addressed in our paper: we could combine the physically-based description of the Richards equation with machine-learning based hydraulic functions that change continuously with season or with the drainage rate. Such an approach would provide insight in the changing hydraulic functions and test the validity of the Richards equation (to see if other processes like macropore flow must be included). We will test this in the future but did not address it in the revised manuscript.

**REVIEW 2 (our reply in blue, the relevant changes made in the manuscript in red)**

General Comments

This manuscript presents an approach for predicting soil water potential and its hysteresis under natural
field conditions by combining deep neural networks (DNN) with autoencoder neural networks. This
integration leverages the strengths of both methods, with the autoencoder effectively compressing and
capturing site-specific features of soil moisture dynamics, and the DNN utilizing these features to
enhance prediction accuracy.

Overall, the method is promising and convincing, and the manuscript is well-organized and clearly
written. I have only a few concerns and suggestions, primarily regarding the model's generalization
capability to clay soils and regions with significantly different climatic conditions, and the model's
interpretability.

We acknowledge the detailed comments of the reviewer. The suggestions and concerns are addressed
below.

**Specific Comments**

**Lines 106-109: Generalization Capability**: Autoencoders are highly dependent on the quality and
diversity of the training data. As shown in Figure 1, the selected region has relatively similar climatic
conditions and soil types, mainly loams with a clay fraction less than 50%. I am curious about the
model's generalization capability to different regions with varying climatic conditions and soil types,
especially for clayey soils. Suggest expanding Section 4.1 to discuss this point and potential approaches
to address this issue. Additionally, suggest discussing the possibility of using other autoencoders, such
as variational autoencoder (VAE).

We agree with the reviewer that the sites presented in the paper are very similar with respect to climate
but also with respect to geology due to their vicinity to the Jura. The "transfer" of the model to sites
with different climate or soil properties will be tested in a following study. But motivated by the
comment of the reviewer, we checked the application of the model for a site called 'Wasen' in the hilly
region of the Napf mountain in the Prealps of Switzerland. In that region the geology is different, and
the climate is wetter and temperatures in winter are lower compared to the sites of the paper. The model
was able to predict the matric potential with high quality, as shown in the figure below. The NSE for
the model was 0.81 indicating that the model performs well in predicting unseen sites in other locations.
We relate the good model performance to the Autoencoder value (AUV=2.54) that was within the range
of the sites presented in the paper (1.9 to 7.0) and we hypothesize that if the autoencoder value deduced
from the water content time series is within the range of the training data, the model performs well.

A paragraph (between quotation below) has been added to Section 4.1 to clarify the model limitations described here:

"For the set of sites analysed in this study, the model showed good generalization capacity and stability.

However, the nine sites were similar with respect to climate and geology and the range of soil textural classes (see Figure 1) was relatively narrow. In a future study, the AUC approach will be applied for sites differing in climate and soil textural classes. We expect that the model can predict the dynamic matric potential for a new site as long as the autoencoder value falls within the range of AUV of the training sites. To predict the soil moisture dynamics for soils with autoencoder values outside of the range of training data, the model must be re-built using additional training data."

[Figure]

*Figure 1. Evaluation of the Deep Neural Network with Autoencoder (AUC-DNN) model performance*

*at the Wasen site for the period 2019-2023. (a) Comparison between the expected Soil Water*

*characteristics curve (SWC) and the observed SWC. (b) Scatter plot that compares observed data points*

*with their corresponding simulated values, providing a visual representation of the level of conformity*

*to the identity line. The two dashed lines represent the 95% confidence interval around the identity line,*

*providing a visual assessment of the level of agreement. (c) Time series comparison showing the*

*observed and predicted matric potential for the entire period. (d) Analysis of the distribution of*

*prediction errors (observed minus modelled value) using positively mild skewed distribution.*

We are grateful for the comment on the variational autoencoder. This could be especially helpful for soils with high variations in water content for the same matric potential value as can be expected for clay soils (the autoencoder values were higher for the four sites with clay content $\geq$ 30%). We expect that using a variational autoencoder instead of a deterministic autoencoder would improve the prediction of matric potential because it leverages regularization in the latent space that explicitly considers the variance (second moment) of the data distribution, leading to a more robust and accurate representation of the water content timeseries (Xu & Liang, 2021; https://doi.org/10.1002/wat2.1533).

A paragraph has been added to Section 4.2 to explain why it is recommended to use the variational autoencoder in case of clay soil (shown below).

"Accordingly, there is no simple interpretation of AUV based on texture and average water content, but the dynamic variation of water content must be considered as well. Due to the relevance of the variation in water content for similar matric potential value, the use of a variational autoencoder (VAE) instead of the typical autoencoder could be considered. In contrast to the typical autoencoder that maps the input information into a single point (or a few points), the VAE produces a probability distribution capturing the variability (second moment) of the data. This could be specifically of interest for clay soils with high water contents (much larger than the residual water content) for the entire range of matric potential values. By including a probabilistic approach in the compressing and decompressing step, the variability of the data could be captured more efficiently using VAR."

**Lines 218: Model Interpretability**: The interpretability of the autoencoder's hidden layer representations is typically challenging. Suggest Including a discussion in the results analysis or discussion section on potential techniques to visualize the features learned by the autoencoder's hidden layers, which can help readers understand the model's internal workings

The interpretation of the autoencoder representation was shortly discussed in section 4.2 and is now expanded to clarify the link between the autoencoder value and the water. The analysis indicates that the value of AUC in the model is not equal to the average water content but is highly affected by it (the higher the average water content, the higher the autoencoder value). Deterministic autoencoders, which map inputs deterministically to a lower-dimensional space, tend to capture prominent statistical properties of the input data. The first moment, or the average, is a primary statistical property. Therefore, the hidden layer representations (AUV) in a deterministic autoencoder will indeed be influenced by the average (first moment) of the water content.

However, the average water content alone cannot explain the distribution of autoencoder values found for the nine sites, but the variations of water content must be included as well. This was shown in section 4.2, revealing that the shape of the envelope embracing all variations in water content must be included to explain the autoencoder value of the nine sites. To confirm that the average water content is not sufficient to classify the dynamics at different sites. First, we run the autoencoder model to analyze the yearly changes. Second, we scale the values from 0 to 1, assigning a value of 0 to the lowest yearly AUV across all sites and a value of 1 to the highest. Finally, we calculate the average yearly water content and scale it similarly to AUV. To highlight the results here, we quantified the annual changes of AUV and average water content for two sites that have Type 2 category (see type description in Figure 6 in the manuscript). AUV tracked changes in average water content but exhibited a different magnitude of variation. The site with higher sand content (Bellach) showed a higher variation of AUV compared to the other site (Zunzgen). This observation supports our conclusion in section 4.2 that the hidden layer is capturing more than just the average water content.

[Figure]

*Figure 2: Annual Variations in Autoencoder Values and Average Water Content for two Sites. The x-axis represents the years, while the y-axis shows the ratio of the scaled AUV to the scaled average water content for the same year. The plot demonstrates how the autoencoder's hidden layer representations track changes in average water content, reflecting variations and additional properties derived from the water content time series.*

To conclude, we consider average water content as a central parameter for visualizing AUV. However, the variability and other higher-order statistical moments (e.g., variance, skewness) significantly influence the precise value of these hidden representations. These additional properties could include:

• Variability (second moment): Reflecting how much the water content fluctuates around the mean.

• Trend: Long-term increase or decrease in water content over time.

• Periodic Components: Seasonal or cyclical patterns in water content.

Therefore, AUV primarily reflects the average water content, combined with other properties derived from the variation in the water content time series. This is discussed in section 4.2.

The text in section 4.2 was edited in multiple locations to address this and the final response was as shown below:

"Simpler models with less parameters could not reproduce the AUV of all sites. Despite the positive correlation between AUV and average water content, the average water content alone is not sufficient to explain the range of AUV for all sites. Also combining average water content with soil texture information could not reproduce the AUVs of all sites, indicating that the soil moisture dynamics represented by AUV is not only dependent on static soil textural attributes but seasonal structural features as well.

Accordingly, there is no simple interpretation of AUV based on texture and average water content, but the dynamic variation of water content must be considered as well."

**Lines 289-290:** Why adopt an NSE value > 0.80 as the criterion for an optimal model? Please provide the rationale for selecting this value.

Several studies have shown that the performance of hydrological modeling is good when NSE values are around 0.75 or higher (Lin et al., 2017; https://doi.org/10.1061/(asce)he.1943-5584.0001580). Other studies suggest categorizing NSE results into levels to evaluate model simulation outcomes, where an

NSE > 0.75 indicates a very good model, while an NSE value < 0.5 signifies unsatisfactory results (Moriasi et al., 2007; https://doi.org/10.13031/2013.23153). In Gupta et al. (1999; https://ascelibrary.org/doi/abs/10.1061/(ASCE)1084-0699(1999)4:2(135)), an NSE value of > 0.80 was considered as good ('efficient') and NSE < 0.50 as poor. These references are now added in the paper.

Here we use NSE > 0.80 as well as criterion for good model performance. For this study, the chosen sites are mainly part of a network designed to provide real-time matric potential information for mitigating soil compaction. We found that when the NSE value is over 0.80, the confidence intervals for matric potential predictions are as follows: around 70 cm for 68% confidence interval, around 120

cm for 90% confidence interval, and around 150 cm for 95% confidence interval. This indicates that despite high predictive accuracy, different confidence intervals provide varying levels of precision and certainty, which can be strategically used for effective soil compaction management:

- 68% Confidence Interval (around ±70 cm): This high precision interval is useful for routine monitoring and precise irrigation adjustments, ensuring that matric potential levels are optimal to prevent over-compaction or drying.
- 90% Confidence Interval (around ±120 cm): This balanced interval offers a reliable estimate for planning soil management practices and designing traffic patterns to minimize soil compaction, providing a good compromise between precision and confidence.
- 95% Confidence Interval (around ±180 cm): This interval, offering the highest confidence, is essential for high-risk scenarios and long-term planning. It ensures that comprehensive measures are in place to prevent severe compaction and maintain soil stability, considering the widest range of potential matric potential variations.

By linking these confidence intervals with high NSE values, we can optimize soil compaction mitigation strategies, tailoring interventions to match the precision and risk tolerance required for various applications, from routine monitoring to high-stakes infrastructure planning.

Two references were added to section 2.4 in the manuscript to address why we use this value.

"A NSE value > 0.75 indicates a very good model, while an NSE value < 0.5 signifies unsatisfactory results (Moriasi et al., 2007). In Gupta et al. (1999) a threshold NSE-value of 0.80 was used for good model performance and is applied here as well."

Figure 2: The common unit for matric potential is -kPa. Please explain the relationship between the -kPa and -cm used in this manuscript.

The data downloaded from the soil moisture network were given in centibars (cbar) with 1 cbar = 1 kPa units of pressure, i.e., energy per volume). In the paper we expressed it as a head (length; energy per weight) considering water density of 1000 kg m$^3$ and gravity acceleration of 10 m s$^{-2}$, resulting in units of cm that are 1/10 of kPa.

This is now stated in section 2.1 as shown below:

"The matric potential in the downloaded data was given in kPa and was transferred to matric potential head with units of cm (1 cm is 0.1 kPa), considering a water density of 1000 kg m$^{-3}$ and gravity acceleration of 10 m s$^{-2}$."

Also, a sentence was added to figure 2:

"The unit of matric potential, represented as -cm, is equivalent to -0.1 kPa."

Equation 1: Please ensure that all parameters are clearly defined after the equation, and that their mathematical notation (bold, italic) is consistent throughout the manuscript.

We rearranged the text to define the parameters after the equations and checked the notation throughout the manuscript.

---

## Author Response (AR2)

**Compare Results**

| Old File: | | New File: |
|---|---|---|
| **egusphere-2024-407-manuscript-version2.pdf** | versus | **22-07-final.pdf** |
| 34 pages (1.80 MB) | | 33 pages (1.70 MB) |
| 01/07/2024 12:12:06 | | 23/07/2024 06:43:05 |

**Total Changes**

**170**

**Content**

Replacements

Insertions

Deletions

**Styling and Annotations**

Styling

Annotations

Go to First Change (page 2)

[revised manuscript text omitted]